# Human Breast Milk: A Source of Potential Probiotic Candidates

**DOI:** 10.3390/microorganisms10071279

**Published:** 2022-06-23

**Authors:** Margherita D’Alessandro, Carola Parolin, Silvia Patrignani, Gilda Sottile, Patrizio Antonazzo, Beatrice Vitali, Rosalba Lanciotti, Francesca Patrignani

**Affiliations:** 1Department of Agricultural and Food Sciences, University of Bologna, 47521 Cesena, Italy; margheri.dalessandr3@unibo.it (M.D.); rosalba.lanciotti@unibo.it (R.L.); 2Department of Pharmacy and Biotechnology, University of Bologna, 40127 Bologna, Italy; carola.parolin@unibo.it (C.P.); b.vitali@unibo.it (B.V.); 3Department of Obstetrics and Gynecology, Maurizio Bufalini Hospital, 47521 Cesena, Italy; silvia.patrignani@auslromagna.it (S.P.); gilda.sottile@auslromagna.it (G.S.); patrizio.antonazzo@auslromagna.it (P.A.); 4Interdepartmental Center for Industrial Agri-Food Research, University of Bologna, 47521 Cesena, Italy

**Keywords:** breast milk, lactobacilli, bifidobacteria, probiotics

## Abstract

This study focuses on the isolation of lactobacilli/bifidobacteria from human breast milk and their first characterization, in the perspective to find new probiotic candidates to be included in food products. More specifically, breast-milk-isolated strains demonstrated a very good aptitude to adhere to intestinal cells, in comparison with *L. rhamnosus* GG strain, taken as reference. The same behavior has been found for hydrophobicity/auto-aggregation properties. A remarkable antagonistic activity was detected for these isolates not only against spoilage and pathogenic species of food interest, but also against the principal etiological agents of intestinal infections. Indeed, isolated strains impaired spoilage and pathogenic species growth, as well as biofilm formation by gut pathogens. In addition, breast milk strains were characterized for their antibiotic susceptibility, displaying species-specific and strain-specific susceptibility patterns. Finally, to assess their technological potential, the fermentation kinetics and viability of breast milk strains in pasteurized milk were investigated, also including the study of the volatile molecule profiles. In this regard, all the strains pointed out the release of aroma compounds frequently associated with the sensory quality of several dairy products such as acetic acid, diacetyl, acetoin, acetaldehyde. Data here reported point up the high potential of breast-milk-isolated strains as probiotics.

## 1. Introduction

Human breast milk, generally recognized as a unique and complex food matrix, could be qualified as an ideal example of a natural functional food [1]. In fact, it contains not only nutrients, hormones, growth factors, immunoglobulins, cytokines, and enzymes, which contribute towards child well-being, but also a significant number of microorganisms. It has been estimated that human breast milk bears 10^3^ CFU/mL of bacteria [2] and represents an important inoculum for the development of the infant gut microbiota, along with skin, mouth, and vaginal tract of the mother [3]. Indeed, it has been well established that the human milk microbiota drives the colonization of the gastrointestinal tract for the newborns, also contributing to the maturation of the immune system [4]. The origin of human milk’s bacteria is still controversial; both facultative anaerobic and aerobic species have been retrieved by molecular and cultural methods. Several authors [1,5,6] have highlighted up to 700 bacterial species in human breast milk, although for an individual, a smaller range (2–18) of cultivable species was reported. *Streptococcus* and *Staphylococcus* were reported not only as the bacterial genera most frequently isolated by human milk, but also as the most abundant, along with *Bifidobacterium* and *Lactobacillus* (and related genera); the latter mainly shape the developing of gut microbiota [3]. Thus, it is not without reason that selected strains of lactic acid bacteria and bifidobacteria, also isolated from breast milk, are worldwide the most investigated microbial species and besides the most used in food and pharma as probiotics. In this instance, however, it is important to underline how breast milk can be considered an important biological niche for the isolation of potential probiotic bacteria, although the technological aptitude of isolates obtained from this matrix is not always guaranteed and needs to be deeply investigated.

Therefore, the aim of this work is directed to the isolation of lactobacilli and bifidobacteria from breast milk samples collected in Italy, focusing on isolates with functional, antimicrobial, and technological potential as probiotics, also in perspective to include them in functional food products. More specifically, specific functional parameters such as hydrophobicity and auto-aggregation and adhesion to a human intestinal cell line were studied for these isolates. In addition, they were evaluated for their antagonistic activity against the pathogenic and spoilage species frequently associated with food products and against the principal etiological agents of intestinal infections. Moreover, these strains were characterized for their anti-biofilm activity and antibiotic susceptibility. Finally, the fermentation kinetics and viability of these lactobacilli during the refrigerated storage in milk were characterized, as well as the volatile molecule profiles of the obtained fermented milks.

## 2. Materials and Methods

### 2.1. Isolation of Bacteria from Breast Milk

As regards to the collection of human breast milk, 30 mothers attending M. Bufalini Hospital in Cesena (Italy) have donated the samples, with full knowledge and written consent about their use. All volunteers provided a written informed consent in accordance with the Ethics Committee of the University of Bologna (Prot. n. 16617, 26 January 2021) and the Ethics Committee of the hospital Maurizio Bufalini (Prot. n. 1523, 12 May 2021) and the institutional review board approved the study. The mothers were interviewed on the kind of delivery (vaginal or by caesarean section, full-term or premature) and on the consumption habits of probiotics. Mothers who received antibiotics or consumed probiotic products during pregnancy or after delivery were excluded.

Sample collection and plating were performed according to [7]. More specifically, to assure the quality of sample collection, all the samples were aseptically collected in sterile tubes and stored on ice until delivery to the laboratory. Breast milk samples were obtained by manual expression after cleaning the nipples and areola with sterile water and discarding the first drops.

Colonies presenting typical lactobacilli or bifidobacteria morphology were isolated and purified. Presumptive lactobacilli or bifidobacteria isolates were stored frozen in MRS broth (Oxoid Ltd., Basingstoke, UK) with 0.05% L-cysteine in glycerol at −80 °C for further studies.

### 2.2. Identification of Isolates

Total DNA of the isolates was obtained from cultures grown in overnight using the commercial GenElute-Bacterial Genomic DNA kit (Sigma, St. Louis, MO, USA) according to the manufacturer’s instructions. All the samples of purified DNA were stored at −20 °C until use. The taxonomic identity of isolates was investigated by amplifying, sequencing and comparing their 16S rRNA gene. Primer forward 27 and primer reverse 1392 were chosen to obtain an amplified fragment of 16S rRNA.

According to the following conditions: 5 min in at 95 °C, 25 cycles of 1 min at 95 °C, 1 min at 55°C and 1.30 min at 72 °C, and a final step of 5 min at 72 °C the amplifications were performed using a PCR System (Applied Biosystems, Foster City, CA, USA). The PCR products were separated on 0.9% (*w*/*v*) agarose gels in TAE buffer, stained with GelRed (Biotium, Hayward, CA, USA) and visualised under UV light (Sambrook and Russell 2001). Amplicons were purified and then their nucleotide sequences were determined by an external service (Eurofins Genomics, Ebersberg, Germany). The identity of the isolates was checked by nucleotide-nucleotide BLAST of the NCBI database (www.ncbi.nlm.nhi.gov/blast (accessed on 3 November 2021)).

### 2.3. Microbial Strains

For the present study, 10 *Lactiplantibacillus*, 4 *Lactobacillus* and 2 *Bifidobacterium* strains were used (Table 1). All these strains are part of the microbial culture collection of DISTAL (Department of Food science and Biotechnology, University of Bologna, Italy). Each strain was isolated from a breast milk sample different from the other samples. These strains were also compared with *Lacticaseibacillus rhamnosus* GG ATCC^®^ 53103™ and *Bifidobacterium animalis* subsp. *lactis* BB-12^®^ used as references. Lactobacilli and bifidobacteria were cultured in MRS broth (Oxoid Ltd., Basingstoke, UK) with 0.05% L-cysteine and incubated at 37 °C for 24 h in anaerobiosis (GasPak System; Oxoid Ltd., Basingstoke, UK). Moreover, in order to evaluate the potential antagonistic activity of the breast milk isolates, selected food spoilage strains, such as *Listeria monocytogenes* ATCC 13932 and Scott A, *Listeria innocua* ATCC 51742, *Enterococcus faecium* BC104, *Escherichia coli* 555, *Staphylococcus aureus* DSM 20231, *Salmonella enteritidis* E5 and MB1409, and the intestinal pathogens enterotoxigenic *E. coli* H10407, *Salmonella choleraesuis* serovar *typhimurium*, *Yersinia enterocolitica* [8], were chosen as target bacterial strains. The pathogenic and spoilage strains were cultured in Brain Heart Infusion (BHI) broth (Oxoid Ltd.) at 37 °C for 24 h.

### 2.4. Hydrophobicity

The bacterial hydrophobicity, as the ability to adhere to hydrocarbons, was evaluated in agreement with [9,10]. Each fresh strain (grown in 24 h of incubation at 37 °C in MRS broth with 0.05% L-cysteine adopting anaerobic conditions) was harvested in the stationary phase after centrifugation at 6000 rpm for 10 min. After removing the supernatant, an isotonic solution of NaCl 0.9% were used to resuspend the microbial pellet and to subsequently dilute the absorbance value to 1, measuring at 560 nm using a spectrophotometer (model 6705, Jenway, Stone, UK). Then, 3 mL of the microbial suspension were vortexed with 0.6 mL of n-hexadecane (Sigma-Aldrich, Milan, Italy) for 4 min. The two phases were left to separate for 1 h at 37 °C. The aqueous phase was removed, and then the absorbance at 560 nm was measured. The hydrophobicity percentage was calculated according to the following formula: (A0 − At)/A0 × 100, where A0 represents the absorbance at time 0 and At represents the absorbance after 1 h of incubation at 37 °C.

### 2.5. Auto-Aggregation Assay

According to the method proposed by [11] and modified by [12], the microbial auto-aggregation for each strain was investigated. Each fresh bacterial culture (grown for 24 h) was centrifuged at 6000 rpm for 10 min. After removing the supernatant, an isotonic solution of NaCl 0.9% was used to resuspend the microbial pellet to the original volume. The auto-aggregation assay was determined during 5 h of incubation at room temperature. More specifically, every hour, 0.1 mL of the upper suspension was taken and placed in a 0.9 mL NaCl 0.9% isotonic solution, and the absorbance (A) was recorded at 600 nm using a spectrophotometer (model 6705, Jenway, Stone, UK). Finally, according to the formula: 1 − (At/A0) × 100, where At represents the mean of absorbance values at time t = 1, 2, 3, 4, or 5 h, and A0 the absorbance at t = 0; the obtained auto-aggregation value was expressed as a percentage.

### 2.6. Adhesion to Differentiated Caco-2 Cells

The capability of breast milk strains to adhere to differentiated Caco-2 intestinal cells was evaluated as described previously [9]. Briefly, Caco-2 cells were grown on glass coverslip in DMEM supplemented with 10% fetal bovine serum, 1% L-glutamine in 5% CO_2_ at 37 °C for 21 days to allow differentiation. Breast-milk-isolated lactobacilli/bifidobacteria were cultured overnight, then subcultured for an additional 18 h. Caco-2 cells were incubated with lactobacilli/bifidobacteria cells by applying a 1:400 ratio at 37 °C with 5% CO_2_ for 3 h, then washed twice with PBS to remove non-adherent lactobacilli. Samples were fixed with methanol and stained with Giemsa, bacterial cells adhering to Caco-2 were counted by at a light-microscope (1000×), considering at least 200 Caco-2 cells.

### 2.7. Antibiotic Susceptibility

The antibiotic susceptibility of the lactobacilli and bifidobacteria isolated from breast milk was determined in Sensititre^®^ antibiotic plates (Thermo Fisher Scientific AG, Basel, Switzerland) in agreement with the recommendations made by EFSA following the official ISO 10932 method. According to the manufacturer’s instructions, bacterial strains were propagated on MRS with 0.05% L-cysteine agar plates and incubated for 48 h at 37 °C in anaerobiosis. Colonies obtained were then resuspended to reach a turbidity of 1 McFarland. Bacterial suspensions were diluited in MRS with 0.05% L-cysteine in order to reach a concentration 10^5^ CFU/mL for the inoculum. A volume of inoculum of 100 µL was then transferred in each well of precoated Sensititre^®^ antibiotic plates and later the plates were incubated for 48 h, at 37 °C, adopting anaerobic conditions. After the incubation time, it was possible to define the minimal inhibitory concentration (MIC), defined as the concentration of the agent that completely prevented visible microbial growth. The tested antibiotics and the relative ranges of concentrations are the followings: Gentamicin 256-0.5 µg/mL, Kanamycin 1024-2 µg/mL, Streptomycin 256-0.5 µg/mL, Neomycin 64-0.12µg/mL, Tetracycline 64-0.12 µg/mL, Erytromycin 8-0.015 µg/mL, Clindamycin 16-0.03 µg/mL, Chloramphenicol 64-0.12 µg/mL, Ampicillin 16-0.03 µg/mL, Penicillin 16-0.03 µg/mL, Vancomycin 128-0.25 µg/mL, Synercid 8-0.015 µg/mL, Linezolid 16-0.03 µg/mL, Trimethoprim 64-0.12 µg/mL, Ciprofloxacin 128-0.25 µg/mL, Rifampicin 64-0.12 µg/mL.

### 2.8. Antagonistic Activity against Spoilage and Pathogenic Species

In order to evaluate the antagonistic ability of lactobacilli and bifidobacteria strains towards the target strains reported in Section 2.3, the method of [13] was performed, adopting some modifications. Fresh cultures of the lactobacilli/bifidobacteria (5 µL) were poured over the surface of MRS + L-cysteine plates (with 1.2% of agar) and let to grow adopting anaerobic conditions at 37 °C for 24 h. Then, 0.1 mL (corresponding to approx. 7 log CFU) of an overnight culture of the target strain was inoculated into 10 mL of BHI soft agar (with 0.7% of agar) and poured on the plates where the lactobacilli/bifidobacteria had grown. After the incubation time (37 °C, 24 h) the plates were checked for the potential presence of the inhibition zone. Then, the inhibition halos were measured from the outer perimeter of the spots in four directions, considering the average diameters. According to the diameters of inhibition, the antagonistic activity showed by all the tested strains was expressed as: −, no inhibition; +, diameter between 1 and 3 mm; ++, diameter between 3 and 6 mm; + + +, diameter between 6 and 10 mm; + + ++, diameter >10 mm.

### 2.9. Inhibition of Development of Intestinal Pathogens Biofilms

Anti-biofilm activity of lactobacilli/bifidobacteria was evaluated as previously described [14], with some modifications. Lactobacilli/bifidobacteria were cultured in MRS medium overnight, then inoculated at a concentration of 6 log CFU/mL and allowed to grow for 24 h. Then, culture supernatant was recovered by centrifugation (2750× *g*, 10 min) and filtered through a 0.22 mm membrane filter to obtain cell-free supernatants. *S. choleraesuis,* enterotoxigenic *E. coli*, *Y. enterocolitica* pathogenic strains were grown in BHI broth at 37 °C overnight, then diluted in the same medium at a concentration of 6 log CFU/mL. 100 μL of pathogen suspension were inoculated in a 96-well culture plate and added with 100 μL of lactobacilli/bifidobacteria cell-free supernatant. Control wells were added with MRS only. The plates were incubated at 37 °C for 72 h. Afterwards, biofilms were quantified by crystal violet staining. Adherent pathogen cells were washed twice with PBS, fixed with ethanol, and stained with 0.41% crystal violet (*w*/*v*) in 12% ethanol. After further washing, crystal violet was solubilized in ethanol and the absorbance was measured at 595 nm (EnSpire Multimode Plate Reader, PerkinElmer Inc., Waltham, MA, USA). Inhibition of pathogen biofilm formation was expressed in percentage relative to the control wells, according to the formula: [1 − (As/Ac)] × 100, where As represents the mean of absorbance values of the sample wells and Ac the absorbance of the control wells.

### 2.10. Fermentation Kinetics in Milk and Viability at Refrigerated Storage

Fresh cultures of lactobacilli and bifidobacteria were inoculated in 50 mL of pasteurized whole milk, in order to reach, as initial concentration, at least 7 log CFU on mL of food matrix. After the inoculum, the milk was incubated at 37 °C for 48 h and in the meantime the acidification kinetics were assessed using a pH meter basic 20 (CRIASON, Modena, Italy). Moreover, the viability of each strain in the fermented milk was performed after 24 h at 37 °C by plating on MRS + 0.05% L-cysteine. In addition, the viability of these strain in the same food matrix was evaluated also considering the adoption of refrigerated conditions (4 °C), that are required during the commercial storage of food products. In this case, the viability of each strain was evaluated by plate counting on MRS + 0.05% L-cysteine until the 21st day of refrigerated storage.

### 2.11. Analysis of Volatile Molecule Profiles in Fermented Milk

The volatile molecule profiles of the tested lactobacilli and bifidobacteria in pasteurized whole milk were investigated. More specifically, the samples of fermented milks were collected after 48 h of incubation at 37 °C, adopting the same conditions reported in Section “Fermentation Kinetics in Milk and Viability at Refrigerated Storage”. The aroma profile of each sample was determined using the solid phase microextraction technique combined with gas chromatography and mass spectrometry (GCMS/SPME), according to the method proposed by [15]. The compounds were then identified by using the Agilent Hewlette Packard NIST 98 mass spectral database.

### 2.12. Strain Survival under Simulated GIT Conditions in Milk

In order to evaluate the resistance of the breast milk isolates to a simulated passage through the stomach and duodenum, the method proposed by [9,16] with certain modifications was performed. More precisely, a sample containing UHT bovine milk with a microbial inoculum of 8–9 log CFU/mL was prepared. The sample was then mixed with the same volume of a “saliva–gastric” solution. The saliva–gastric solution contained CaCl_2_ (0.22 g/L), NaCl (16.2 g/L), KCl (2.2 g/L), NaHCO3 (1.2 g/L), and 0.3% (*w*/*v*) porcine pepsin (Sigma). The sample was quickly brought to pH values of 2.5–3 with HCl 1 M and then was moved to a thermostatic bath for 90 min at 37 °C (WB-MF, Falc Instruments, Treviglio, Italy). Afterwards, a specific volume was taken to proceed with the first sampling of the cells’ viability (gastric digestion). In the meantime, 2 mL of the same sample were centrifuged (12,000 rpm, 4 min and 4 °C), the obtained microbial pellet was washed with 2 mL of NaCl 0.9% isotonic solution and then resuspended in 2 mL of bile extract porcine solution (Sigma) at a concentration of 1% in PBS, which simulated the hepatic bile. Then, the sample was moved to a thermostatic bath at 37 °C for 10 min in order to simulate the duodenal shock phase of the bile. Then, 100 µL of the sample was taken for the third sampling in order to verify how this duodenal shock could affect cell viability. The remaining sample was subjected to centrifugation (12,000 rpm for 4 min at 4 °C), the microbial pellet was then resuspended to the initial volume with NaCl 0.9% isotonic solution. After that, a third solution, representing enteric stress (0.3% bile and 0.1% pancreatin from porcine pancreas dissolved in PBS) was added. The last incubation time in the thermostatic bath was 90 min at 37 °C. Then, 100 µL was taken from the sample for the last sampling (intestinal digestion). Samples were plated on MRS agar plates with 0.05% L-cysteine and incubated at 37 °C for 24–48 h in anaerobic conditions.

### 2.13. Statistical Analysis

The microbiological and chemical-physical data are the mean of 3 repetitions. The obtained data were analyzed by Statistica software (version 8.0; StatSoft, Tulsa, OK, USA) adopting the analysis of variance (ANOVA), and Tukey’s test for data comparisons.

## 3. Results and Discussion

### 3.1. Hydrophobicity and Auto-Aggregation

Hydrophobicity and auto-aggregation results are reported in Figure 1 and Figure 2, respectively. Hydrophobicity and auto-aggregation were evaluated also for *L. rhamnosus* GG and *B. animalis* subsp. *lactis* BB-12, that are recognized probiotics and used here as a reference strains. It was observed that breast milk strains showed a different behavior in terms of hydrophobicity (Figure 1). The highest values were detected for *L. gasseri* 32T0A (85.11%), *L. plantarum* 30b6a (78.40%), *B. longum* 32T0B.bis (77.87%). As regards to the other strains *L. plantarum* 32T0C, *L. plantarum* 3.6D, *L. plantarum* 11.3C, *L. plantarum* 31T0C, *L. plantarum* 29T0L, *L. plantarum* 34T0B showed, respectively, hydrophobicity values of 77.59%, 75.47%, 72.18%, 70.73%, 69.28%, 65.82%. The remaining strains expressed hydrophobicity levels below 60%. With regard to the auto-aggregation results, in this case each strain has also expressed a specific behavior (Figure 2). The most promising values were 89.77%, 85.65%, 84.67%, 80.73%, recorded for *B. longum* 32T0 B.bis, *L. gasseri* g.1, *L. plantarum* 31 T0 C, *B. animalis* BL6. In addition, *L. plantarum* 3.6D, *L. plantarum* 30b6A, *L. plantarum* 34T0B, *L. gasseri* CFl11, *L. plantarum* M6C, *L. plantarum* 33.1G, *L. plantarum* 29T0L, *L. gasseri* 32 T0 A, *L. plantarum* 35 T0 B.bis showed a rate of auto-aggregation of 67.80%, 67.25%, 65.82%, 65.24%, 64.01%, 63.15%, 54.40%, 53.40%, 50.50%, whereas the remaining strains indicated percentages below 50%. In this framework, it’s interesting to underline how the hydrophobicity and autoaggregation data observed for the majority of the investigated strains are even more relevant than those showed by GG ATCC^®^53103™ (52.29% hydrophobicity and 34.05% autoaggregation) BB-12^®^ (45.14% hydrophobicity and 37.92% autoaggregation). In this context, the recorded data are useful in order to understand the potential functionality of the different strains. As reported by several authors [9,17,18], the hydrophobic nature of the surface of a microorganism could be associated with a better attachment of the strain to host cells and consequently represents an advantage for its permanence in the gastrointestinal tract. Other studies [19,20] have discussed about the hydrophilic behavior of bacteria, that occurs with values similar or less than 40% whereas the hydrophobic nature is highlighted for those bacteria with average of more than 40%. According to these considerations, almost all of these strains showed ranges up than 40%, with the only exception for CFl11 with 34.90%. As an addition factor to describe the potential ability of these strains to adhere to the gastrointestinal tract, a high autoaggregation capability was detected for the majority of the strains under study. Moreover, an interesting correlation between hydrophobicity and auto-aggregation data was highlighted for selected strains such as *B. longum* 32T0B.bis, *L. plantarum* 31 T0 C, *L. plantarum* 3.6D, which showed higher values for these two investigated parameters.

### 3.2. Adhesion of Breast Milk Strains to Intestinal Cells

As already stated, the ability of a candidate probiotic to adhere to gut mucosal surface is a desired important characteristic, since adhesion process represents the first step of microbial colonization and high adhesiveness ensures for microbial persistence in a given environment. In order to assess the adhesiveness of breast milk lactobacilli/bifidobacteria, differentiated Caco-2 cells were employed, as they resemble significant features of human enterocytes with a brush border layer as found in the small intestine. In this sense, in Table 2, the data related to the adhesive properties of the strains, expressed as the num. of lactobacilli/bifidobacteria cells on Caco-2 cells, are reported. Adhesion values varied between 4.4 and 34.2 bacterial cells/Caco-2 cell, being *L. plantarum* 29T0L, *L. plantarum* 34T0B *and L. gasseri* 34T0C the most adhesive strains. Most breast-milk-isolated strains showed a very good aptitude to adhere to intestinal cells, even higher than *L. rhamnosus* GG (15.6 ± 3.2). This behavior with respect to *L. rhamnosus* GG, a well-known probiotic strain endowed with remarkable beneficial features, has been already underlined for hydrophobicity and auto-aggregation properties, pointing up the high potential of breast-milk-isolated strains as probiotics. As mentioned before, hydrophobic nature of the surface of microorganisms influences their adhesion to intestinal cells, indeed lactobacilli/bifidobacterial strains herein analyzed are characterized by valuable hydrophobicity and ability to adhere to Caco-2 cells. Notably, *L. gasseri* CFl11 strain showed the lowest hydrophobicity and the lowest adhesiveness. *L. gasseri* CFl11 together with *L. gasseri* g1 strain showed high level of auto-aggregation but low adhesion to Caco-2 cells, suggesting that the interaction of microbial cells with different biotic surfaces (i.e., other bacteria or human epithelium) can involve different surface molecules [21].

### 3.3. Antibiotic Susceptibility

In order to better assess the safety of these strains, their antibiotic susceptibility was investigated considering a wide spectrum of antibiotics (Table 3). In fact, also for EFSA, during the selection of starter, co-starter or functional microorganisms the determination of their antibiogram is considered as an important prerequisite [22]. The antibiogram (Table 3) of breast milk lactobacilli and bifidobacteria showed a specific behavior that is related to the species and the strain considered. This behavior is also confirmed by several studies [23,24]. More specifically as regards to our data, the obtained results have reported that among all the considered antibiotics, Clindamycin, Ampicillin, Penicillin, Linezolid, Rifampicin exhibited the highest bactericidal effect against all the selected strains. Otherwise, Gentamicin, Kanamycin, Streptomycin, Neomycin, Vancomycin, Trimethoprim showed lower antibacterial activities, especially for the *Lactobacillus* strains. In this framework, literature data has clearly indicated the natural resistance of different lactic acid bacteria and bifidobacteria strains to several antibiotics [24,25]. Lactobacilli, pediococci and *Leuconostoc* spp. have been reported to express a high natural resistance to Vancomycin. This behaviour could be useful to discern them from other Gram-positive bacteria [26]. Other studies performed by [27,28] were conducted in order to establish the levels of susceptibility of *Lactobacillus* spp. to various antimicrobial agents such as Vancomycin, Kanamycin, Tetracycline, Penicillin, Erythromycin and Chloramphenicol and this sensitivity was shown to be species-dependent. Additionally, during our investigation, not only a species-dependent sensitivity to antibiotics was observed for *L. plantarum* and *L. gasseri,* but also a different behaviour for different strains belonging to the same species. More generally, as regards the variability of susceptibility to antibiotics within the species, among the strains belonging to *L. plantarum*, variability was observed only for gentamicin; whereas the strains belonging to *L. gasseri* showed considerable variability for most of the antibiotics used. As regards to bifidobacteria, as reported by [23,29,30], they are usually very susceptible to Gram-positive spectrum antibiotics (Erythromycin, Lincomycin, Novobiocin, Teicoplanin and Vancomycin), broad-spectrum antibiotics (Rifampicin, and Chloramphenicol) and beta-lactams (Penicillin, Ampicillin, Amoxicillin, Piperacillin, Ticarcillin and Imipenem). In contrast, most *Bifidobacterium* species are generally resistant to Neomycin, Gentamicin, Kanamycin and Streptomycin [29,30], as shown also in the present study (Table 3). Furthermore, considering the potential inclusion of these strains in food products, further insights are necessary to deeply characterize the resistance mechanism.

### 3.4. Antagonistic Activity against Spoilage and Pathogenic Species of Food Interest

The antagonist activities of the breast-milk-isolated strains against several spoilage species were evaluated. As shown in Table 4, all tested strains showed a remarkable inhibitory activity against pathogens of food interest such as *L. monocytogenes* ATCC 13932, *L. monocytogenes* SCOTT A, *L. innocua* ATCC 51742, *S. enteritidis* MB1409, *S. enteritidis* E5, *E. faecium* BC104, *E. coli* 555 and *S. aureus* DSM 20231. More specifically half of the lactobacilli determined inhibition zones ranging between 6 and 10 mm toward *L. monocytogenes* SCOTT A, *L. innocua* ATCC 51742, *S. enteritidis* MB1409, *S. enteritidis* E5, *E. faecium* BC104, *E. coli* 555 and *S. aureus* DSM 20231. Moreover, almost all lactobacilli/bifidobacteria showed an antagonist activity, with inhibition zones ranging between 1 and 6 mm, also against *L. monocytogenes* ATCC 13932. In this framework, although most of these strains have shown a strong antagonist activity, among all the better results were recorded for *L. plantarum* 3.6D, that also against *L. monocytogenes* ATCC 13932 produced an inhibition zone ranging between 6 and 10 mm. In contrast, for *B. longum* 32T0Bbis and *B. animalis* BL6, the lowest antagonistic activity against almost all the target microorganisms under study were recorded. In this framework, *B. longum* 32T0Bbis was the only strain that showed no activity towards a pathogen (i.e., *S. aureus* DSM 20231).

### 3.5. Antagonistic Activity against Intestinal Pathogenic Species

Enterotoxigenic *E. coli*, *S. choleraesuis,* and *Y. enterocolitica* were employed as strains representative of the major pathogens responsible for intestinal infections. The antagonistic activity of breast milk lactobacilli/bifidobacteria was evaluated as described in par 2.6, and results are reported in Table 5. All the isolated strains turned to be highly effective towards intestinal pathogens, with inhibition zones above 6 mm. Most strains showed inhibition zones above 10 mm against *Y. enterocolitica,* with the only exception of *L. plantarum* 29T0L and *L. plantarum* 30b6A. The best anti-pathogen profile was recorded for *L. plantarum* 3.6D, confirming the remarkable antimicrobial activity previously observed towards spoilage strains. The antimicrobial activity was also analyzed in terms of anti-biofilm effect since the sessile mode of growth of microorganisms contribute to their persistence and resistance to biotic and abiotic factors. From the clinical viewpoint, the establishment of a biofilm by a pathogenic species on human mucosae generally increases resistance to antibiotics as well as the persistence over time of the pathogen itself, challenging its eradication. In this perspective, the ability of a beneficial strain to interfere not only with pathogen growth, but also to its adherence and biofilm formation, represents an advantage. Breast-milk-isolated strains all showed the ability to inhibit pathogen biofilm formation, as reported in Figure 3. Overall, *L. plantarum* strains were highly effective towards the three tested pathogens (52.9–100% biofilm inhibition), particularly towards *Y. enterocolitica* (79.8–96% biofilm inhibition). *L. gasseri* strains strongly inhibited enterotoxigenic *E. coli* biofilm formation (59–97.6% biofilm inhibition) but were less active towards *S. choleraesuis* (23.9–66.7% biofilm inhibition). The best anti-profile was registered for *L. plantarum* 33.1G and *L. plantarum* 34T0B (>80% inhibition against all pathogens), followed by *L. plantarum* 32T0C and *L. plantarum* 35T0BBis.

### 3.6. Fermentation Kinetics in Pasteurized Milk and Viability at Refrigerated Storage

In order to better understand the technological potential of these selected breast milk strains, their kinetics of fermentation and viability in milk at 37 °C were studied. As shown in Table 6, the results have indicated for most of these strains slow fermentation kinetics that are not acceptable for dairy industries. These data are useful to underline their unsuitability as fermentation starters. In sight of this, several authors have also highlighted that the probiotic bacteria belonging to *Bifidobacterium* and *Lactobacillus* spp. are generally used as additional cultures, also because of the scarce sensory properties of fermented milk obtained using them as starters [31,32]. Nevertheless, as reported in Table 7, for most of these strains the maintenance of high viability after 24 h of incubation at 37 °C in pasteurized whole milk was observed. After these considerations, in order to better understand their potential use in dairy products, the viability of these strains in pasteurized milk in refrigeration conditions for 21 days was also investigated (Table 8). In this framework, some of the strains, such as *L. plantarum* 3.6D, *L. plantarum* M6C, *L. plantarum* 31T0C, *L. plantarum* 32T0C, showed the maintenance of the highest cells viability until 21 days at 4 °C (>6.5 log CFU/g), indicating their potential suitability as adjunct cultures in a dairy product such as a fermented milk. In fact, from an applicative point of view, as reported by [33,34], in order to develop satisfactory fermented probiotic products, the viable cell count at the moment of consumption should be above 6 log CFU/g in order to fulfill standards proposed by the International Dairy Federation (IDF 1992) and to provide the intake of a sufficient “daily dose” of viable bacteria.

### 3.7. Volatile Molecules Profiles of Inoculated Pasteurized Milks

The analysis of the volatilome of milks inoculated with lactobacilli/bifidobacteria isolated from human breast milk and collected after 48 h of incubation at 37 °C showed a specific pattern of molecules, generally belonging to the chemical classes of ketones, alcohols, aldehydes and acids. In relation to the strain considered, the relative percentages of the detected molecules are reported in Table 9. More specifically, *B. longum* 32T0Bbis and *B. animalis* BL6 were characterized by the highest abundances of acids, in particular of acetic acid. In this sense, the high levels of acetic acid and hexanoic acid identified, especially in the volatilome profiles of the *Bifidobacterium* species, could be involved in their strong antagonistic activity towards the selected target microorganisms. In fact, several studies have investigated the strong antimicrobial activity against Gram-positive and Gram-negative microorganisms showed by selected short-chain organic acids, such as acetic, butanoic and hexanoic due to the release of the acids through the cell membrane of microorganisms [34,35]. More precisely, the undissociated organic acids are able to function as protonophores, inducing the acidification of the cytoplasm and the accumulation of toxic anions. The decrease in the cell’s internal pH affects the influx of protons through the cell membrane, which dissipates the proton-motive force, reducing cellular energy (ATP) and affecting substrate uptake in the cell [36,37]

Differently, the strains belonging to *L. plantarum* and *L. gasseri* produced a limited number of acids, including acetic acid ranging between 3.74% and 15.70% (*L. plantarum*) and 5.63% and 25.77% (*L. gasseri*). As regards to the production of ethanol, only *L. plantarum* 35T0Bbis produced a significative amount of this compound. In addition, all the tested strains revealed the production of diacetyl, acetoin, and also acetaldehyde, considered as molecules that could play a crucial role for the sensory characterization of several dairy products [34,38,39]. In this context, several authors have reported how the microbial ability to release diacetyl and acetoin could be considered as an important feature for the selection of lactic acid bacteria as starters [40]. Moreover, it is important to underline that all of these strains, especially the lactic acid bacteria, showed a remarkable production of acetaldehyde. This molecule is desired and appreciated, as well described by [34,39], since its presence could contribute to specific flavour in yogurt and fermented milks.

### 3.8. Strain Survival under Simulated GIT Conditions in Milk

In order to deeply investigate the overall resistance of the selected breast milk strains inoculated in milk (8–9 log CFU/mL), their survival rate after a simulated digestive process was assessed (Figure 4, Figure 5 and Figure 6). Regarding their resistance during and after the simulated digestion process, the decrease in viability was slight for all strains considered, especially for *L. plantarum* strains. In fact, the majority of these strains showed a survival rate at the end of the simulated process of at least 7 log CFU/mL. In particular, the highest cell viability rates were recorded for *L. plantarum* 29 T0 L, M6C, 30b6A with a final survival level of approx. 8 log CFU/mL. A different behaviour was detected only for *L. plantarum* 11.3 *C* which reached the lowest survival rate (6.53 log CFU/mL) at the end of simulated intestinal phase. As regards to *L. gasseri* strains, a good survival rate was detected especially for *L. gasseri* 34T0C and 32 T0A, as demonstrated by the recorded cell viability of approx. 7 log CFU/mL after the simulated GIT conditions. Regarding instead the bifidobacteria strains, *B. longum* B.Bis showed a final survival rate (approx. 7 log CFU/mL) higher of 1 log compared with *B. animalis* BL6.

## 4. Conclusions

In recent years, human breast milk has proved to be an interesting source for obtaining new and specific probiotic strains, including lactic acid bacteria and bifidobacteria, also for infants, with the aim of promoting their correct immunological and intestinal microbiota development [41,42]. In fact, these bacterial groups have been suggested to play an important role in the reduction in the incidence and severity of infections in breastfed infants [43,44]. In addition, a recent study [41] evaluated the probiotic potential of several bacteria isolated from human breast milk and belonging to lactic acid bacteria e bifidobacteria groups. More in general, some probiotic strains of the genera *Lactobacillus* and *Bifidobacterium* have been widely investigated for their potential resistance to acidic environments, competition against pathogens and immunological properties in vitro and in vivo. Furthermore, these genera, especially *Lactobacillus*, are commonly used as co-starters in the production of several dairy products, exhibiting good viability in low pH products such as fermented milk during both the fermentation process and the refrigerated storage of the product [45].

In this context, our data clearly highlight the functional potential of strains isolated from breast milk. In fact, the majority of the strains employed in the present study showed a remarkable aptitude to adhere to intestinal cells, even higher than *L. rhamnosus* GG, a recognized probiotic strain. In this sense, it is important to report how the ability of a candidate probiotic to adhere to gut mucosal surface contributes to the microbial persistence in a specific environment. This behavior resembles hydrophobicity and auto-aggregation features of breast milk strains herein analyzed, underlining their high potential as probiotics. In addition, all tested strains showed a remarkable inhibitory activity against pathogens of food interest as well as intestinal pathogens. More specifically, the majority of the lactobacilli produced halos of inhibitions ranging between 6 and 10 mm toward *L. monocytogenes* SCOTT A, *L. innocua* ATCC 51742, *S. enteritidis* MB1409, *S. enteritidis* E5, *E. faecium* BC104, *E. coli* 555 and *S. aureus* DSM 20231. All the isolated strains turned to be highly effective also towards intestinal pathogens, especially against *Y. enterocolitica*. The antimicrobial activity was also analyzed in terms of anti-biofilm effect, since, from a clinical point of view, the establishment of a biofilm by a pathogenic species increases antibiotic resistance and makes its eradication challenging. Thus, the ability of a beneficial strain to interfere with pathogen adherence and biofilm formation represents an advantage. Additionally, in this case, all breast-milk-isolated strains showed the capability to inhibit the formation of biofilm by pathogens, in particular *L. plantarum* strains were very effective towards the tested pathogens. Moreover, in order to also investigate their technological potential, the strain fermentation kinetics and viability in pasteurized whole milk were investigated, along with the analysis of the volatile molecule profiles of the fermented milks. The results clearly indicated the unsuitability as fermentation starters for the majority of the strains, due to their slow fermentation kinetics. Nevertheless, especially for some strains, the maintenance of high viability in pasteurized milk has been highlighted, also during the refrigerated storage. In this framework, *L. plantarum* 3.6D, *L. plantarum* M6C, *L. plantarum* 31T0C, *L. plantarum* 32T0C showed the best cell viability profile until 21 days at 4°C (>6.5 log CFU/g), indicating their potential suitability as adjunct cultures in a dairy product such as a fermented milk. Their potential role as co-starters was confirmed especially for the lactic acid bacteria since the volatilome of milks inoculated with these selected lactobacilli/bifidobacteria showed a release of diacetyl, acetoin, and acetaldehyde, that could positively contribute to the specific flavour of dairy products. Instead, *B. longum* 32T0Bbis and *B. animalis* BL6 strains were characterized by the highest production of organic acids, such as acetic acid, and fatty acids, such as hexanoic one. Such levels of acetic acid could explain their strong antagonistic activity against the selected target microorganisms. Moreover, with regard to antibiotic susceptibility, our results are in agreement with literature data reporting intrinsic resistance to a wide range of antibiotics, especially for lactic acid bacteria. In these cases, further studies are needed to better characterize mechanisms responsible for antibiotic resistance, before including these strains in food products. In conclusion, the attained data could represent an important contribution to better understanding the proper application of the studied strains, also considering their final use. In this context, the selection of the most promising strains is strongly connected with the final objective to achieve and the relative purposes. However, among the studied strains, *B. animalis* BL6 and *L. gasseri* 34 T0C can be considered as the most promising strains to be used in functional products.

## Figures and Tables

**Figure 1 microorganisms-10-01279-f001:**
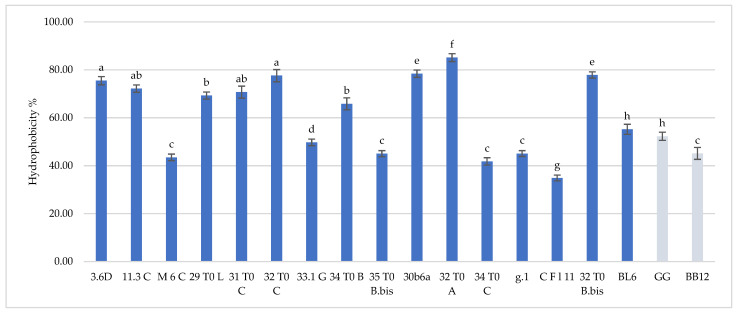
Cell hydrophobicity of *L. plantarum* 3.6D, 11.3 C, M 6 C, 29 T0 L, 31 T0 C, 32 T0 C, 33.1 G, 34 T0 B, 35 T0 B.Bis, 30 b6 A, *L. gasseri* 32T0A, 34 T0C, g.1, C F l11, *B. longum* 32T0B.Bis, *B. animalis* BL6 and *L. rhamnosus* GG and *B. animalis* subsp. *lactis* BB-12. Results are reported as average ± SD. Samples equipped with different letters are significantly different (*p* < 0.05).

**Figure 2 microorganisms-10-01279-f002:**
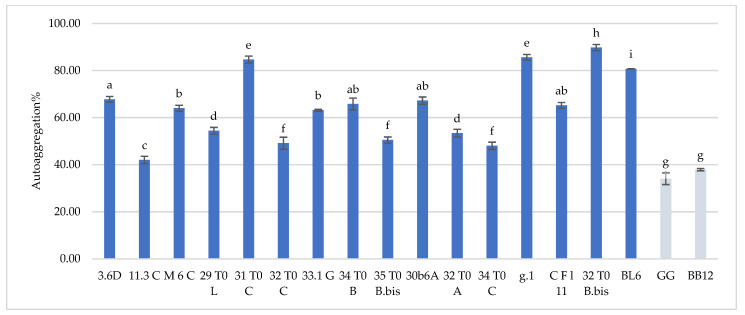
Cell autoaggregation of *L. plantarum* 3.6D, 11.3 C, M 6 C, 29 T0 L, 31 T0 C, 32 T0 C, 33.1 G, 34 T0 B, 35 T0 B.Bis, 30 b6 A, *L. gasseri* 32T0A, 34 T0C, g.1, C F l11, *B. longum* 32T0B.Bis, *B. animalis* BL6 and *L. rhamnosus* GG and *B. animalis* subsp. *lactis* BB-12. Results are reported as average ± SD. Samples equipped with different letters are significantly different (*p* < 0.05).

**Figure 3 microorganisms-10-01279-f003:**
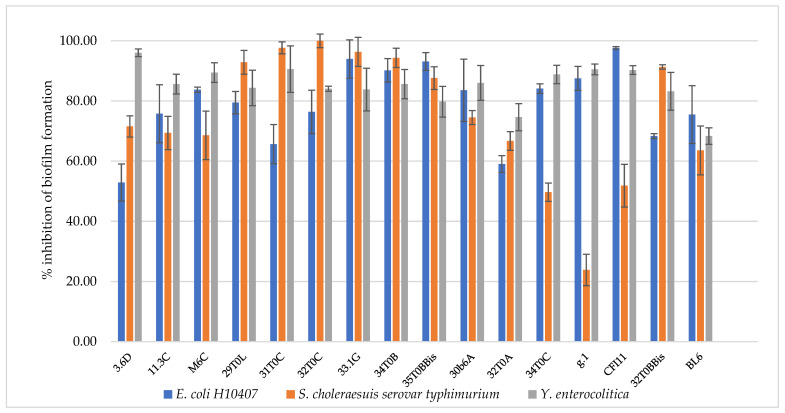
Inhibition of pathogen biofilm formation by *L. plantarum* 3.6D, 11.3 C, M 6 C, 29 T0 L, 31 T0 C, 32 T0 C, 33.1 G, 34 T0 B, 35 T0 B.Bis, 30 b6 A, *L. gasseri* 32T0A, 34 T0C, g.1, C F l11, *B. longum* 32T0B.Bis, *B. animalis* BL6.

**Figure 4 microorganisms-10-01279-f004:**
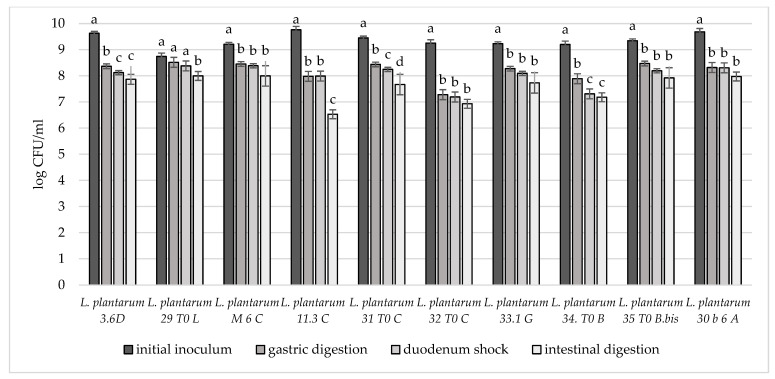
Cell loads of *L. plantarum* 3.6D, 29 T0 L, M 6 C, 11.3 C, 31 T0 C, 32 T0 C, 33.1 G, 34 T0 B, 35 T0 B.Bis, 30 b6 A after the simulated stomach–duodenum passage, performed immediately after the inoculation in milk. Samples with different letters are significant different (*p* < 0.05).

**Figure 5 microorganisms-10-01279-f005:**
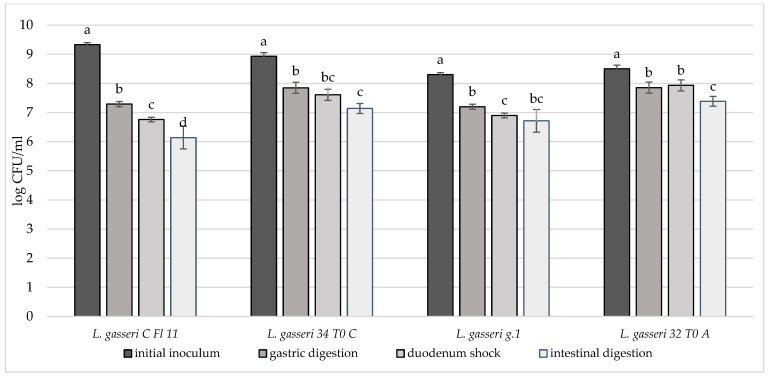
Cell loads of *L. gasseri* CFl11, *L. gasseri* 34T0C, *L. gasseri* g.1, *L. gasseri* 32T0A after the simulated stomach–duodenum passage, performed immediately after the inoculation in milk. Samples with different letters are significantly different (*p* < 0.05).

**Figure 6 microorganisms-10-01279-f006:**
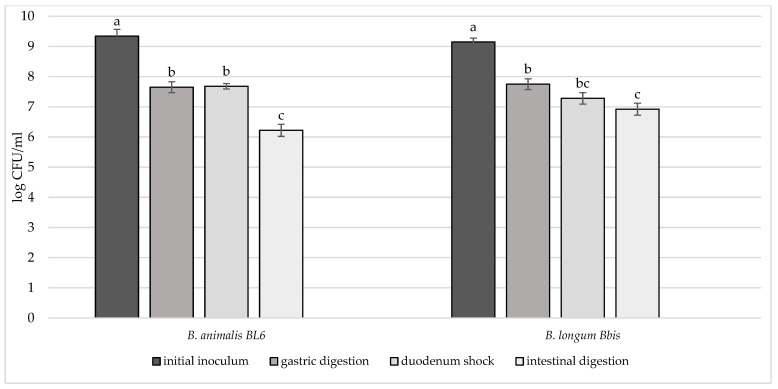
Cell loads of *B. animalis BL6*
*and B. longum B. Bis* after the simulated stomach–duodenum passage, performed immediately after the inoculation in milk. Samples with different letters are significantly different (*p* < 0.05).

**Table 1 microorganisms-10-01279-t001:** Lactiplantibacillus, *Lactobacillus* and *Bifidobacterium* strains isolated from human breast milk and used in the present study.

Strains	Species	Isolation Source	Collection
3.6 D	*L. plantarum*	Breast milk	DISTAL
11.3 C	*L. plantarum*	Breast milk	DISTAL
M 6 C	*L. plantarum*	Breast milk	DISTAL
29 T0 L	*L. plantarum*	Breast milk	DISTAL
31 T0 C	*L. plantarum*	Breast milk	DISTAL
32 T0 C	*L. plantarum*	Breast milk	DISTAL
33.1 G	*L. plantarum*	Breast milk	DISTAL
34 T0 B	*L. plantarum*	Breast milk	DISTAL
35 T0 B.bis	*L. plantarum*	Breast milk	DISTAL
30 b 6 A	*L. plantarum*	Breast milk	DISTAL
32 T0 A	*L. gasseri*	Breast milk	DISTAL
34 T0 C	*L. gasseri*	Breast milk	DISTAL
g.1	*L. gasseri*	Breast milk	DISTAL
C F l 11	*L. gasseri*	Breast milk	DISTAL
32 T0 B.bis	*B. longum*	Breast milk	DISTAL
BL6	*B. animalis*	Breast milk	DISTAL

**Table 2 microorganisms-10-01279-t002:** Adhesion of breast milk lactobacilli/bifidobacteria strains on differentiated Caco-2 cells. Data are expressed as number of adherent microbial cells/Caco-2 cell and shown as average ± SD. Samples equipped with different letters are significantly different (*p* < 0.05).

Strain	Adhesion (n. Adherent Microbes/Caco-2 Cell)
*L. plantarum* 3.6D	22.9 ± 8.6 ^a,c,l^
*L. plantarum* 11.3C	14.3 ± 8.4 ^b,h,m^
*L. plantarum* M6C	25.7 ± 11.2 ^a,g^
*L. plantarum* 29T0L	34.2 ±13.4 ^d^
*L. plantarum* 31T0C	24.7 ± 8.5 ^a,l^
*L. plantarum* 32T0C	21.2 ± 6.8 ^c,f^
*L. plantarum* 33.1 G	20.2 ± 7.7 ^c^
*L. plantarum* 34T0B	28.7 ± 11.3 ^e^
*L. plantarum* 35T0Bbis	17.1 ± 7.9 ^b,f^
*L. plantarum* 30b6A	24.8 ± 9.0 ^a,g,l^
*L. gasseri* 32T0A	14.7 ± 4.0 ^b,h^
*L. gasseri* 34T0C	27.7 ± 12.1 ^e,g^
*L. gasseri* g.1	13.9 ± 6.9 ^h,m^
*L. gasseri* CFl11	4.4 ± 3.7 ^i^
*B. longum* 32T0Bbis	22.5 ± 6.0 ^c,l^
*B. animalis* BL6	13.7 ± 5.4 ^h,m^
*L. rhamnosus* GG	15.6 ± 3.2 ^m^

**Table 3 microorganisms-10-01279-t003:** Evaluation of minimum inhibitory concentrations (MIC, µg/mL) of selected antibiotics against *Lactobacillus* and *Bifidobacterium* strains isolated from breast milk and used in the present study.

Strain	Gentamicin	Kanamycin	Streptomycin	Neomycin	Tetracycline	Erytromycin	Clindamycin	Chloramphenicol	Ampicillin	Penicillin	Vancomycin	Dalfopristin	Linezolid	Trimethoprim	Ciprofloxacin	Rifampicin
*L. plantarum* 3.6D	128	612	128	>64	16	1	4	8	0.25	0.25	>128	8	4	>64	64	1
*L. plantarum* 11.3C	256	1024	>256	>64	16	1	4	4	0.12	0.5	>128	4	4	>64	64	1
*L. plantarum* M6C	128	1024	>256	>64	16	1	2	4	0.25	0.5	>128	4	4	>64	64	1
*L. plantarum* 29T0L	256	>1024	>256	>64	16	2	4	8	0.12	0.5	>128	4	4	>64	64	2
*L. plantarum* 31T0C	>256	1024	>256	>64	16	2	4	8	0.12	0.5	>128	4	4	>64	64	2
*L. plantarum* 32T0C	64	1024	>256	>64	16	2	8	8	0.12	0.5	>128	2	2	>64	32	1
*L. plantarum* 33.1 G	256	>1024	>256	>64	16	2	4	8	0.12	0.5	>128	4	4	>64	64	8
*L. plantarum* 34T0B	256	>1024	>256	>64	16	2	8	8	0.12	0.5	>128	2	2	>64	32	2
*L. plantarum* 35T0Bbis	256	>1024	>256	>64	16	2	4	8	0.12	1	>128	4	2	>64	32	1
*L. plantarum* 30b6A	64	1024	>256	>64	16	1	4	4	0.1	0.5	>128	4	2	>64	64	1
*L. gasseri* 32T0A	64	1024	>256	>64	16	1	4	8	0.25	0.06	2	1	2	8	32	0.12
*L. gasseri* 34T0C	256	1024	>256	>64	32	2	8	8	0.12	0.5	>128	2	2	>64	64	1
*L. gasseri* g.1	128	1024	128	>64	4	0.5	1	4	0.25	0.06	2	1	1	16	32	0.12
*L. gasseri* CFl11	32	256	8	>64	2	0.12	0.6	4	0.12	0.06	2	1	1	16	32	0.12
*B. longum* 32T0Bbis	32	1024	128	>64	1	0.03	0.03	0.5	0.12	0.06	0.5	0.06	0.12	4	4	0.12
*B. animalis* BL6	256	1024	128	>64	32	0.12	0.06	1	0.06	0.12	1	0.25	0.5	0.12	8	0.25

**Table 4 microorganisms-10-01279-t004:** Evaluation of the antagonistic activity of the *Lactobacillus* and *Bifidobacterium* strains isolated from breast milk and used in the present study against selected pathogenic or spoilage microorganisms related to foods.

Strain	*L. monocytogenes* ATCC 13932	*L. monocytogenes* SCOTT A	*L. innocua* ATCC 51742	*S. enteritidis* MB1409	*S. enteritidis* E5	*E. faecium* BC104	*E. coli* 555	*S. aureus* DSM 20231
*L. plantarum* 3.6D	+++	+++	+++	+++	+++	+++	++++	+++
*L. plantarum* 11.3C	++	+++	+++	+++	+++	+++	+++	+++
*L. plantarum* M6C	++	++	+++	+++	+++	+++	+++	+++
*L. plantarum* 29T0L	++	++	++++	+++	+++	+++	+++	+++
*L. plantarum* 31T0C	++	++	+++	+++	+++	+++	+++	+++
*L. plantarum* 32T0C	++	++	++++	++++	+++	+++	++	++++
*L. plantarum* 33.1 G	++	+++	+++	+++	+++	+++	+++	+++
*L. plantarum* 34T0B	++	+++	+++	+++	+++	+++	+++	+++
*L. plantarum* 35T0Bbis	++	+	+++	+++	+++	+++	+++	+++
*L. plantarum* 30b6A	++	+++	++	+++	++++	+++	+++	+++
*L. gasseri* 32T0A	++	+++	++	+++	+++	++	++	+++
*L. gasseri* 34T0C	++	+++	+++	+++	+++	+++	+++	+++
*L. gasseri* g.1	++	++	+	++	++	++	+++	+++
*L. gasseri* CFl11	++	++	++	++	++	++	+++	++
*B. longum* 32T0Bbis	+	+	+	++	+	+	+	-
*B. animalis* BL6	+	++	+	+	++	++	++	+

Legend: −, no inhibition; +, inhibition 1–3 mm; ++, inhibition 3–6 mm; +++, inhibition 6–10 mm; ++++, >10 mm. The diameter of inhibition, for each strain, was the average of three replicates.

**Table 5 microorganisms-10-01279-t005:** Evaluation of the antagonistic activity of the *Lactobacillus* and *Bifidobacterium* strains isolated from breast milk and used in the present study against intestinal pathogenic species.

Strain	*E. coli* H10407	*S. choleraesuis serovar typhimurium*	*Y. enterocolitica*
*L. plantarum* 3.6D	++++	++++	++++
*L. plantarum* 11.3C	++++	+++	++++
*L. plantarum* M6C	+++	+++	++++
*L. plantarum* 29T0L	+++	+++	+++
*L. plantarum* 31T0C	+++	+++	++++
*L. plantarum* 32T0C	+++	+++	++++
*L. plantarum* 33.1 G	++++	+++	++++
*L. plantarum* 34T0B	++++	+++	++++
*L. plantarum* 35T0Bbis	+++	+++	++++
*L. plantarum* 30b6A	++++	+++	+++
*L. gasseri* 32T0A	++++	+++	++++
*L. gasseri* 34T0C	++++	+++	++++
*L. gasseri* g.1	+++	+++	++++
*L. gasseri* CFl11	+++	+++	++++
*B. longum* 32T0Bbis	+++	++++	++++
*B. animalis* BL6	+++	+++	++++

Legend: −, no inhibition; +, inhibition 1–3 mm; ++, inhibition 3–6 mm; +++, inhibition 6–10 mm; ++++, >10 mm. The diameter of inhibition, for each strain, was the average of three replicates.

**Table 6 microorganisms-10-01279-t006:** Kinetics of acidifications (reported as changes of pH values) of lactobacilli and bifidobacteria inoculated in pasteurized whole milk. The sampling points are expressed as hours. The pH values are the average of three replicates with a variability <5%.

Strain	t0	t3	t6	t11	t14	t16	t20	t24	t29	t31	t34	t37	t42	t48
*L. plantarum* 3.6D	6.53	6.45	6.39	6.28	6.21	6.15	6.11	6.07	5.81	5.67	5.58	5.29	5.28	5.17
*L. plantarum* 11.3C	6.50	6.42	6.27	6.17	6.11	6.03	5.97	5.95	5.88	5.78	5.69	5.61	5.35	5.04
*L. plantarum* M6C	6.60	6.51	6.28	6.23	6.21	6.08	6.13	5.91	5.78	5.35	5.13	4.82	4.63	4.09
*L. plantarum* 29T0L	6.57	6.40	6.25	6.22	6.02	5.87	5.77	5.68	5.62	5.51	5.44	5.31	5.25	5.02
*L. plantarum* 31T0C	6.57	6.54	6.44	6.35	6.16	6.04	5.87	5.79	5.76	5.64	5.53	5.30	5.26	5.14
*L. plantarum* 32T0C	6.56	6.51	6.36	6.31	6.06	5.95	5.79	5.72	5.61	5.59	5.51	5.27	5.25	5.10
*L. plantarum* 33.1 G	6.55	6.49	6.34	6.27	6.10	6.01	5.84	5.67	5.68	5.60	5.50	5.31	5.30	5.13
*L. plantarum* 34T0B	6.56	6.48	6.39	6.32	6.26	6.16	6.11	6.05	6.25	6.24	6.03	5.91	5.40	5.19
*L. plantarum* 35T0Bbis	6.59	6.53	6.47	6.39	6.36	6.34	6.30	5.96	5.78	5.31	5.24	5.22	5.13	5.05
*L. plantarum* 30b6A	6.59	6.52	6.41	6.34	6.23	6.21	6.09	6.07	6.05	6.02	5.92	5.76	5.68	5.80
*L. gasseri* 32T0A	6.55	6.51	6.43	6.36	6.18	6.05	5.89	5.78	5.77	5.63	5.45	5.33	5.26	5.05
*L. gasseri* 34T0C	6.62	6.54	6.39	6.32	6.23	6.09	6.11	6.07	6.04	6.01	5.97	5.94	5.92	5.88
*L. gasseri* g.1	6.56	6.50	6.35	6.33	6.27	6.13	6.02	5.91	5.84	5.79	5.67	5.45	5.26	5.24
*L. gasseri* CFl11	6.63	6.55	6.39	6.31	6.24	6.11	6.07	6.02	5.98	5.94	5.89	5.84	5.77	5.71
*B. longum* 32T0Bbis	6.58	6.40	6.16	6.21	6.04	5.96	5.88	5.79	5.68	5.67	5.55	5.48	5.46	5.38
*B. animalis* BL6	6.48	6.43	6.39	6.31	6.27	6.18	6.12	6.03	5.98	5.92	5.88	5.81	5.76	5.69

**Table 7 microorganisms-10-01279-t007:** Microbial viability of the lactobacilli and bifidobacteria under study in pasteurized whole milk (log CFU/mL) after inoculation (t0) and after 24 h (t24) of incubation at 37 °C. The cell counts are expressed as the average of three replicates; the values are reported ± standard deviations (SD).

Strain	t0	t24
*L. plantarum* 3.6D	7.32 ± 0.23	7.49 ± 0.13
*L. plantarum* 11.3C	6.12 ± 0.13	7.32 ± 0.15
*L. plantarum* M6C	7.89 ± 0.19	8.77 ± 0.18
*L. plantarum* 29T0L	6.70 ± 0.17	7.76 ± 0.18
*L. plantarum* 31T0C	7.15 ± 0.23	8.6 ± 0.16
*L. plantarum* 32T0C	7.2 ± 0.16	9.25 ± 0.25
*L. plantarum* 33.1 G	6.95 ± 0.13	9.3 ± 0.25
*L. plantarum* 34T0B	7.5 ± 0.23	7.88 ± 0.19
*L. plantarum* 35T0Bbis	6.69 ± 0.17	7.39 ± 0.21
*L. plantarum* 30b6A	6.93 ± 0.21	7.55 ± 0.24
*L. gasseri* 32T0A	7.12 ± 0.13	7.98 ± 0.19
*L. gasseri* 34T0C	6.99 ± 0.17	7.39 ± 0.21
*L. gasseri* g.1	6.94 ± 0.21	7.95 ± 0.24
*L. gasseri* CFl11	6.74 ± 0.21	7.97 ± 0.22
*B. longum* 32T0Bbis	7.23 ± 0.19	8.18 ± 0.19
*B. animalis* BL6	7.17 ± 0.18	8.88 ± 0.09

**Table 8 microorganisms-10-01279-t008:** Microbial viability of the lactobacilli and bifidobacteria under study inoculated in pasteurized whole milk (log CFU/mL) after 14 (t14), 21 (t21) days of refrigerated storage (4 °C). The cell counts are expressed as the average of three replicates; the values are reported ± standard deviations (SD).

Strain	t7	t14	t21
*L. plantarum* 3.6D	7.53 ± 0.13	7.14 ± 0.15	6.89 ± 0.16
*L. plantarum* 11.3C	7.22 ± 0.11	7.08 ± 0.23	6.32 ± 0.17
*L. plantarum* M6C	7.57 ± 0.29	7.16 ± 0.08	6.69 ± 0.15
*L. plantarum* 29T0L	7.60 ± 0.12	7.24 ± 0.12	6.06 ± 0.14
*L. plantarum* 31T0C	7.75 ± 0.13	7.48 ± 0.17	6.93 ± 0.15
*L. plantarum* 32T0C	7.92 ± 0.14	7.25 ± 0.15	6.85 ± 0.24
*L. plantarum* 33.1 G	7.95 ± 0.33	7.43 ± 0.15	6.32 ± 0.45
*L. plantarum* 34T0B	7.41 ± 0.23	6.97 ± 0.19	5.88 ± 0.29
*L. plantarum* 35T0Bbis	7.29 ± 0.16	7.08 ± 0.11	5.69 ± 0.12
*L. plantarum* 30b6A	6.99 ± 0.11	7.15 ± 0.14	6.05 ± 0.24
*L. gasseri* 32T0A	7.64 ± 0.13	6.48 ± 0.19	6.07 ± 0.19
*L. gasseri* 34T0C	8.02 ± 0.17	6.88 ± 0.21	6.39 ± 0.11
*L. gasseri* g.1	7.84 ± 0.11	6.15 ± 0.21	5.25 ± 0.27
*L. gasseri* CFl11	7.27 ± 0.21	5.92 ± 0.12	5.47 ± 0.12
*B. longum* 32T0Bbis	7.94 ± 0.21	5.95 ± 0.14	5.45 ± 0.14
*B. animalis* BL6	7.89 ± 0.11	6.9 ± 0.22	6.17 ± 0.12

**Table 9 microorganisms-10-01279-t009:** Volatile compounds (reported as relative percentages) of pasteurized whole milk samples, inoculated with lactobacilli/bifidobacteria isolated from human breast milk and collected after 48 h of incubation at 37 °C; detected by GC-MS-SPME technique.

	*L. plantarum*										*L. gasseri*				*B. longum B. animalis*	
	3.6D	11.3C	M6C	29T0L	31T0C	32T0C	331G	34T0B	35T0Bbis	30b6A	32T0A	34T0C	G.1	CFl11	32T0Bbis	BL6
Acetone	9.90	11.15	12.07	15.11	11.18	12.28	9.96	12.34	11.41	10.74	10.46	8.00	10.83	8.91	3.08	2.71
Cyclopentanone	2.40	3.03	2.53	4.05	2.04	2.52	2.32	2.37	2.24	1.98	2.27	1.48	1.67	2.73	0.81	0.73
2-Butanone	5.81	6.15	6.33	7.38	5.83	6.51	6.41	6.56	6.41	6.02	6.17	3.70	5.86	4.98	1.44	1.34
Diacetyl	0.87	0.26	0.29	0.46	0.18	0.12	0.23	0.81	0.33	0.22	0.50	0.91	1.97	1.78	0.70	1.84
2-Pentanone	9.49	10.36	11.30	12.38	9.57	11.55	12.09	12.71	8.65	11.42	11.03	6.07	9.92	9.23	2.76	2.63
Isobutenyl ketone	3.19	2.97	3.88	6.98	3.33	2.76	2.62	1.54	3.30	2.53	2.56	3.36	2.59	2.49	1.59	1.82
2-Hexanone	2.11	2.48	1.81	11.92	1.35	1.69	4.77	1.41	1.57	4.68	1.87	3.55	1.82	1.17	0.77	0.92
2 Heptanone	32.39	37.32	38.98	3.19	33.14	35.16	38.39	38.23	25.95	37.74	35.70	28.72	31.90	34.97	12.12	10.44
Acetoin	0.31	0.22	0.22	0.66	0.40	0.29	0.47	0.32	1.31	0.48	1.73	0.21	0.27	1.64	0.11	0.16
2-Nonanone	7.51	8.36	9.10	11.67	7.11	7.84	8.24	8.24	7.55	9.35	7.76	9.37	6.51	8.80	3.82	3.00
**Total ketones**	**73.98**	**82.31**	**86.50**	**73.80**	**74.13**	**80.71**	**85.50**	**84.53**	**68.73**	**85.17**	**80.06**	**65.39**	**73.34**	**76.72**	**27.21**	**25.59**
1-Butanol	0.48	0.91	0.42	1.08	0.08	0.22	0.44	0.48	0.41	0.65	0.25	0.59	0.37	1.15	0.57	0.39
3-Hexanol-	3.53	3.56	3.86	2.88	3.46	2.90	3.89	3.02	4.36	2.83	3.17	3.79	3.78	3.22	1.76	1.75
1-Pentanol	0.39	2.91	0.39	1.04	4.03	4.30	0.81	3.44	4.86	4.01	4.40	0.33	0.30	6.77	2.58	0.22
1-Octanol	0.38	0.33	0.34	0.80	0.27	0.28	0.39	0.36	0.34	0.46	0.46	0.36	0.25	0.52	0.18	0.16
Ethanol	0.67	2.25	0.42	1.89	0.53	0.81	0.40	0.78	4.83	0.46	0.38	0.84	0.66	0.69	0.60	0.46
**Total alcohols**	**5.46**	**9.96**	**5.43**	**7.69**	**8.37**	**8.50**	**5.93**	**8.09**	**14.81**	**8.42**	**8.66**	**5.92**	**5.36**	**12.36**	**5.69**	**2.98**
Acetaldehyde	4.86	3.80	4.33	5.25	11.89	2.78	3.64	3.49	5.37	1.06	5.65	2.92	4.36	2.97	3.83	17.08
**Total aldehydes**	**4.86**	**3.80**	**4.33**	**5.25**	**11.89**	**2.78**	**3.64**	**3.49**	**5.37**	**1.06**	**5.65**	**2.92**	**4.36**	**2.97**	**3.83**	**17.08**
Acetic acid	9.64	2.29	1.52	6.66	3.30	4.21	2.38	1.57	5.62	2.49	2.86	14.75	10.19	3.73	51.18	47.59
Hexanoic acid	3.12	1.18	0.82	3.41	1.24	2.21	1.07	0.93	2.49	1.09	1.23	5.75	2.47	1.35	6.45	4.00
Butanoic acid	0.29	0.28	0.51	0.71	0.29	0.25	0.52	0.52	0.32	0.65	0.56	0.58	1.40	0.59	0.64	0.30
Octanoic acid	2.65	0.18	0.89	2.47	0.79	1.33	0.97	0.88	2.66	1.12	0.98	4.70	2.87	2.28	4.99	2.46
**Total acids**	**15.70**	**3.94**	**3.74**	**13.25**	**5.61**	**8.01**	**4.93**	**3.89**	**11.09**	**5.35**	**5.63**	**25.77**	**16.93**	**7.96**	**63.27**	**54.36**

## Data Availability

Not applicable.

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
