# Peer review of "Human Breast Milk: A Source of Potential Probiotic Candidates"

_microorganisms, 2022, doi:10.3390/microorganisms10071279_

Round 1

Reviewer 1 Report

Although the article titled “Human breast milk: a source of potential probiotic candidates” is intriguing, the method is a little old one and the description of the manuscript appears not so clear-cut. 

1)    According to the title “Human breast milk: a source of potential probiotic candidates”, it is suggested that the bacteria isolated from the milk are actually colonizing the milk-secreting epithelial or intralobular duct cells in the breast or suspended in the milk. However, the majority of actually isolated strains are L. plantarum (Table 1), which has been recognized as habitants on the plants. Therefore, those isolates might be contaminants from the surface skin around nipples. Nevertheless, there is no enough description in the Materials and Methods as to how the human breast milk was sampled without risk of such contamination. 

2)    Even though a probiotic strain is derived from the human breast milk, that must be active in the intestine, a site with different environment from human breast milk, in order to work as a probiotic. What advantage is expected by screening those probiotics for the gut from bacteria adapting themselves to the human breast milk?

3)    To work as probiotics in the intestine, probiotic strains should survive gastric acid, bile acid and pancreatic enzyme after they are orally taken. There are no such experiments to examine the resistance of the isolates to those hazardous agents in the article. 

4)    Adhesion ability to the intestinal mucosa has been considered crucial for a strain to be an efficient probiotic.  In order to estimate the attachment property, the authors examined cell hydrophobicity (Fig. 1), cell auto-aggregation (Fig. 2) and adhesion to Caco-2 gut epithelial cell line (Table 2). Whereas 29TOL, 34TOB and 34TOC are the top 3 for adhesion ability to Caco-2, those strains are not the top 3 anymore in hydrophobicity or auto-aggregation. How do authors think the dissociation between them?

5)    Authors examined suppressive effect of isolates on spoilage and pathogenic bacteria (Tables 4 and 5). What substances in the culture supernatant are expected to be responsible for the suppressive effect? Was there any correlation between the suppressive effect and the concentration of short-chain fatty acids?

6)    After all, which isolate(s) do authors recommend for the best probiotic in their study?       

Author Response

Dear Editor in Chief,

We would like to thank the Referee 1 for the careful revision on the submitted manuscript.  We addressed all the suggestions and we modified the manuscript according to the revision (changes are highlighted in red) also reducing the overlapping index (changes are highlighted in blue). In our opinion, the manuscript was improved, and some interesting trails were added (Strain survival under simulated GIT conditions in milk).

Thanking you for your support

Best Regards

Francesca Patrignani

Reviewer 2 Report

The article is devoted to the characteristics of 14 strains of lactobacilli and 2 strains of bifidobacteria isolated from breast milk. The properties of the strains that are important for their use as potential probiotics have been studied. The research methods are described in detail, the results are presented in 9 tables and 3 figures, and there is no doubt about their reliability. The article is well written and edited. Objections are raised by discussions of the results. Section “4. Conclusion” partially repeats the text from section ”3. Results and Discussion”. The “Results and Discussion” section should be divided into “Results” and “Discussins”, the “Discussion” combining the fragments from sections 3. and 4. removing repetitions. Information should be provided on similar works on bacterial strains from breast milk. The “Conclusion” section should contain a brief summary of the importance of bacterial strains from breast milk and a description of the most promising strain out of 16 studied.

Separate remarks:

1. The variability of antibiotic susceptibility within species should be noted. For L.pl. variability is noted only for gentamicin; for L. gasseri there is considerable variability for most of the antibiotics used.

2. Lanes 335-336, 452-453.

“the majority of the lactobacilli determined inhibition zones ranging between 6 and 10 mm towards L. monocytogenes SCOTT A…” - 7 strains out of 14 (Table 4) are not the majority.

3.Lanes 358-359.

“The best anti-pathogen profile was recorded for L. plantarum 30b6A” - According to Table 5, this is rather a strain of L. plantarum 3.6D.

The article is devoted to the characteristics of 14 strains of lactobacilli and 2 strains of bifidobacteria isolated from breast milk. The properties of the strains that are important for their use as potential probiotics have been studied. The research methods are described in detail, the results are presented in 9 tables and 3 figures, and there is no doubt about their reliability. The article is well written and edited. Objections are raised by discussions of the results. Section “4. Conclusion” partially repeats the text from section ”3. Results and Discussion”. The “Results and Discussion” section should be divided into “Results” and “Discussins”, the “Discussion” combining the fragments from sections 3. and 4. removing repetitions. Information should be provided on similar works on bacterial strains from breast milk. The “Conclusion” section should contain a brief summary of the importance of bacterial strains from breast milk and a description of the most promising strain out of 16 studied.

Separate remarks:

1. The variability of antibiotic susceptibility within species should be noted. For L.pl. variability is noted only for gentamicin; for L. gasseri there is considerable variability for most of the antibiotics used.

2. Lanes 335-336, 452-453.

“the majority of the lactobacilli determined inhibition zones ranging between 6 and 10 mm towards L. monocytogenes SCOTT A…” - 7 strains out of 14 (Table 4) are not the majority.

3.Lanes 358-359.

“The best anti-pathogen profile was recorded for L. plantarum 30b6A” - According to Table 5, this is rather a strain of L. plantarum 3.6D.

The article is devoted to the characteristics of 14 strains of lactobacilli and 2 strains of bifidobacteria isolated from breast milk. The properties of the strains that are important for their use as potential probiotics have been studied. The research methods are described in detail, the results are presented in 9 tables and 3 figures, and there is no doubt about their reliability. The article is well written and edited. Objections are raised by discussions of the results. Section “4. Conclusion” partially repeats the text from section ”3. Results and Discussion”. The “Results and Discussion” section should be divided into “Results” and “Discussins”, the “Discussion” combining the fragments from sections 3. and 4. removing repetitions. Information should be provided on similar works on bacterial strains from breast milk. The “Conclusion” section should contain a brief summary of the importance of bacterial strains from breast milk and a description of the most promising strain out of 16 studied.

Separate remarks:

1. The variability of antibiotic susceptibility within species should be noted. For L.pl. variability is noted only for gentamicin; for L. gasseri there is considerable variability for most of the antibiotics used.

2. Lanes 335-336, 452-453.

“the majority of the lactobacilli determined inhibition zones ranging between 6 and 10 mm towards L. monocytogenes SCOTT A…” - 7 strains out of 14 (Table 4) are not the majority.

3.Lanes 358-359.

“The best anti-pathogen profile was recorded for L. plantarum 30b6A” - According to Table 5, this is rather a strain of L. plantarum 3.6D.

The article is devoted to the characteristics of 14 strains of lactobacilli and 2 strains of bifidobacteria isolated from breast milk. The properties of the strains that are important for their use as potential probiotics have been studied. The research methods are described in detail, the results are presented in 9 tables and 3 figures, and there is no doubt about their reliability. The article is well written and edited. Objections are raised by discussions of the results. Section “4. Conclusion” partially repeats the text from section ”3. Results and Discussion”. The “Results and Discussion” section should be divided into “Results” and “Discussins”, the “Discussion” combining the fragments from sections 3. and 4. removing repetitions. Information should be provided on similar works on bacterial strains from breast milk. The “Conclusion” section should contain a brief summary of the importance of bacterial strains from breast milk and a description of the most promising strain out of 16 studied.

Separate remarks:

1. The variability of antibiotic susceptibility within species should be noted. For L.pl. variability is noted only for gentamicin; for L. gasseri there is considerable variability for most of the antibiotics used.

2. Lanes 335-336, 452-453.

“the majority of the lactobacilli determined inhibition zones ranging between 6 and 10 mm towards L. monocytogenes SCOTT A…” - 7 strains out of 14 (Table 4) are not the majority.

3.Lanes 358-359.

“The best anti-pathogen profile was recorded for L. plantarum 30b6A” - According to Table 5, this is rather a strain of L. plantarum 3.6D.

The article is devoted to the characteristics of 14 strains of lactobacilli and 2 strains of bifidobacteria isolated from breast milk. The properties of the strains that are important for their use as potential probiotics have been studied. The research methods are described in detail, the results are presented in 9 tables and 3 figures, and there is no doubt about their reliability. The article is well written and edited. Objections are raised by discussions of the results. Section “4. Conclusion” partially repeats the text from section ”3. Results and Discussion”. The “Results and Discussion” section should be divided into “Results” and “Discussins”, the “Discussion” combining the fragments from sections 3. and 4. removing repetitions. Information should be provided on similar works on bacterial strains from breast milk. The “Conclusion” section should contain a brief summary of the importance of bacterial strains from breast milk and a description of the most promising strain out of 16 studied.

Separate remarks:

1. The variability of antibiotic susceptibility within species should be noted. For L.pl. variability is noted only for gentamicin; for L. gasseri there is considerable variability for most of the antibiotics used.

2. Lanes 335-336, 452-453.

“the majority of the lactobacilli determined inhibition zones ranging between 6 and 10 mm towards L. monocytogenes SCOTT A…” - 7 strains out of 14 (Table 4) are not the majority.

3.Lanes 358-359.

“The best anti-pathogen profile was recorded for L. plantarum 30b6A” - According to Table 5, this is rather a strain of L. plantarum 3.6D.

The article is devoted to the characteristics of 14 strains of lactobacilli and 2 strains of bifidobacteria isolated from breast milk. The properties of the strains that are important for their use as potential probiotics have been studied. The research methods are described in detail, the results are presented in 9 tables and 3 figures, and there is no doubt about their reliability. The article is well written and edited. Objections are raised by discussions of the results. Section “4. Conclusion” partially repeats the text from section ”3. Results and Discussion”. The “Results and Discussion” section should be divided into “Results” and “Discussins”, the “Discussion” combining the fragments from sections 3. and 4. removing repetitions. Information should be provided on similar works on bacterial strains from breast milk. The “Conclusion” section should contain a brief summary of the importance of bacterial strains from breast milk and a description of the most promising strain out of 16 studied.

Separate remarks:

1. The variability of antibiotic susceptibility within species should be noted. For L.pl. variability is noted only for gentamicin; for L. gasseri there is considerable variability for most of the antibiotics used.

2. Lanes 335-336, 452-453.

“the majority of the lactobacilli determined inhibition zones ranging between 6 and 10 mm towards L. monocytogenes SCOTT A…” - 7 strains out of 14 (Table 4) are not the majority.

3.Lanes 358-359.

“The best anti-pathogen profile was recorded for L. plantarum 30b6A” - According to Table 5, this is rather a strain of L. plantarum 3.6D.

The article is devoted to the characteristics of 14 strains of lactobacilli and 2 strains of bifidobacteria isolated from breast milk. The properties of the strains that are important for their use as potential probiotics have been studied. The research methods are described in detail, the results are presented in 9 tables and 3 figures, and there is no doubt about their reliability. The article is well written and edited. Objections are raised by discussions of the results. Section “4. Conclusion” partially repeats the text from section ”3. Results and Discussion”. The “Results and Discussion” section should be divided into “Results” and “Discussins”, the “Discussion” combining the fragments from sections 3. and 4. removing repetitions. Information should be provided on similar works on bacterial strains from breast milk. The “Conclusion” section should contain a brief summary of the importance of bacterial strains from breast milk and a description of the most promising strain out of 16 studied.

Separate remarks:

1. The variability of antibiotic susceptibility within species should be noted. For L.pl. variability is noted only for gentamicin; for L. gasseri there is considerable variability for most of the antibiotics used.

2. Lanes 335-336, 452-453.

“the majority of the lactobacilli determined inhibition zones ranging between 6 and 10 mm towards L. monocytogenes SCOTT A…” - 7 strains out of 14 (Table 4) are not the majority.

3.Lanes 358-359.

“The best anti-pathogen profile was recorded for L. plantarum 30b6A” - According to Table 5, this is rather a strain of L. plantarum 3.6D.

The article is devoted to the characteristics of 14 strains of lactobacilli and 2 strains of bifidobacteria isolated from breast milk. The properties of the strains that are important for their use as potential probiotics have been studied. The research methods are described in detail, the results are presented in 9 tables and 3 figures, and there is no doubt about their reliability. The article is well written and edited. Objections are raised by discussions of the results. Section “4. Conclusion” partially repeats the text from section ”3. Results and Discussion”. The “Results and Discussion” section should be divided into “Results” and “Discussins”, the “Discussion” combining the fragments from sections 3. and 4. removing repetitions. Information should be provided on similar works on bacterial strains from breast milk. The “Conclusion” section should contain a brief summary of the importance of bacterial strains from breast milk and a description of the most promising strain out of 16 studied.

Separate remarks:

1. The variability of antibiotic susceptibility within species should be noted. For L.pl. variability is noted only for gentamicin; for L. gasseri there is considerable variability for most of the antibiotics used.

2. Lanes 335-336, 452-453.

“the majority of the lactobacilli determined inhibition zones ranging between 6 and 10 mm towards L. monocytogenes SCOTT A…” - 7 strains out of 14 (Table 4) are not the majority.

3.Lanes 358-359.

“The best anti-pathogen profile was recorded for L. plantarum 30b6A” - According to Table 5, this is rather a strain of L. plantarum 3.6D.

The article is devoted to the characteristics of 14 strains of lactobacilli and 2 strains of bifidobacteria isolated from breast milk. The properties of the strains that are important for their use as potential probiotics have been studied. The research methods are described in detail, the results are presented in 9 tables and 3 figures, and there is no doubt about their reliability. The article is well written and edited. Objections are raised by discussions of the results. Section “4. Conclusion” partially repeats the text from section ”3. Results and Discussion”. The “Results and Discussion” section should be divided into “Results” and “Discussins”, the “Discussion” combining the fragments from sections 3. and 4. removing repetitions. Information should be provided on similar works on bacterial strains from breast milk. The “Conclusion” section should contain a brief summary of the importance of bacterial strains from breast milk and a description of the most promising strain out of 16 studied.

Separate remarks:

1. The variability of antibiotic susceptibility within species should be noted. For L.pl. variability is noted only for gentamicin; for L. gasseri there is considerable variability for most of the antibiotics used.

2. Lanes 335-336, 452-453.

“the majority of the lactobacilli determined inhibition zones ranging between 6 and 10 mm towards L. monocytogenes SCOTT A…” - 7 strains out of 14 (Table 4) are not the majority.

3.Lanes 358-359.

“The best anti-pathogen profile was recorded for L. plantarum 30b6A” - According to Table 5, this is rather a strain of L. plantarum 3.6D.

The article is devoted to the characteristics of 14 strains of lactobacilli and 2 strains of bifidobacteria isolated from breast milk. The properties of the strains that are important for their use as potential probiotics have been studied. The research methods are described in detail, the results are presented in 9 tables and 3 figures, and there is no doubt about their reliability. The article is well written and edited. Objections are raised by discussions of the results. Section “4. Conclusion” partially repeats the text from section ”3. Results and Discussion”. The “Results and Discussion” section should be divided into “Results” and “Discussins”, the “Discussion” combining the fragments from sections 3. and 4. removing repetitions. Information should be provided on similar works on bacterial strains from breast milk. The “Conclusion” section should contain a brief summary of the importance of bacterial strains from breast milk and a description of the most promising strain out of 16 studied.

Separate remarks:

1. The variability of antibiotic susceptibility within species should be noted. For L.pl. variability is noted only for gentamicin; for L. gasseri there is considerable variability for most of the antibiotics used.

2. Lanes 335-336, 452-453.

“the majority of the lactobacilli determined inhibition zones ranging between 6 and 10 mm towards L. monocytogenes SCOTT A…” - 7 strains out of 14 (Table 4) are not the majority.

3.Lanes 358-359.

“The best anti-pathogen profile was recorded for L. plantarum 30b6A” - According to Table 5, this is rather a strain of L. plantarum 3.6D.

The article is devoted to the characteristics of 14 strains of lactobacilli and 2 strains of bifidobacteria isolated from breast milk. The properties of the strains that are important for their use as potential probiotics have been studied. The research methods are described in detail, the results are presented in 9 tables and 3 figures, and there is no doubt about their reliability. The article is well written and edited. Objections are raised by discussions of the results. Section “4. Conclusion” partially repeats the text from section ”3. Results and Discussion”. The “Results and Discussion” section should be divided into “Results” and “Discussins”, the “Discussion” combining the fragments from sections 3. and 4. removing repetitions. Information should be provided on similar works on bacterial strains from breast milk. The “Conclusion” section should contain a brief summary of the importance of bacterial strains from breast milk and a description of the most promising strain out of 16 studied.

Separate remarks:

1. The variability of antibiotic susceptibility within species should be noted. For L.pl. variability is noted only for gentamicin; for L. gasseri there is considerable variability for most of the antibiotics used.

2. Lanes 335-336, 452-453.

“the majority of the lactobacilli determined inhibition zones ranging between 6 and 10 mm towards L. monocytogenes SCOTT A…” - 7 strains out of 14 (Table 4) are not the majority.

3.Lanes 358-359.

“The best anti-pathogen profile was recorded for L. plantarum 30b6A” - According to Table 5, this is rather a strain of L. plantarum 3.6D.

The article is devoted to the characteristics of 14 strains of lactobacilli and 2 strains of bifidobacteria isolated from breast milk. The properties of the strains that are important for their use as potential probiotics have been studied. The research methods are described in detail, the results are presented in 9 tables and 3 figures, and there is no doubt about their reliability. The article is well written and edited. Objections are raised by discussions of the results. Section “4. Conclusion” partially repeats the text from section ”3. Results and Discussion”. The “Results and Discussion” section should be divided into “Results” and “Discussins”, the “Discussion” combining the fragments from sections 3. and 4. removing repetitions. Information should be provided on similar works on bacterial strains from breast milk. The “Conclusion” section should contain a brief summary of the importance of bacterial strains from breast milk and a description of the most promising strain out of 16 studied.

Separate remarks:

1. The variability of antibiotic susceptibility within species should be noted. For L.pl. variability is noted only for gentamicin; for L. gasseri there is considerable variability for most of the antibiotics used.

2. Lanes 335-336, 452-453.

“the majority of the lactobacilli determined inhibition zones ranging between 6 and 10 mm towards L. monocytogenes SCOTT A…” - 7 strains out of 14 (Table 4) are not the majority.

3.Lanes 358-359.

“The best anti-pathogen profile was recorded for L. plantarum 30b6A” - According to Table 5, this is rather a strain of L. plantarum 3.6D.

Author Response

Dear Editor in Chief,

We would like to thank the Referee 2 for the careful revision on the submitted manuscript.  We addressed all the suggestions and we modified the manuscript according to the revision (changes are highlighted in red), also reducing the overlapping index (changes are highlighted in blue). In our opinion, the manuscript was improved, and some interesting trails were added (Strain survival under simulated GIT conditions in milk).

Thanking you for your support

Best Regards

Francesca Patrignani

Round 2

Reviewer 1 Report

The revised manuscript are well done, and the replies by authors are fully enough to solve the reviewer's query.